# Evaluation of Overshunting between Low and Medium Pressure Ventriculoperitoneal Shunts in Dogs with Severe Hydrocephalus Using Frameless Stereotactic Ventricular Shunt Placement

**DOI:** 10.3390/ani13121890

**Published:** 2023-06-06

**Authors:** Kanokwan Keadwut, Pakthorn Lewchalermwong, Nathanat Inpithuk, Piyathip Choochalermporn, Ananya Pongpradit, Nattika Koatsang, Nirut Suwanna

**Affiliations:** 1Kasetsart University Veterinary Teaching Hospital, Faculty of Veterinary Medicine, Kasetsart University, Bangkok 10900, Thailand; kanokwan.ke@ku.th (K.K.);; 2Department of Companion Animal Clinical Sciences, Faculty of Veterinary Medicine, Kasetsart University, Bangkok 10900, Thailand

**Keywords:** dog, hydrocephalus, overshunting, frameless stereotactic, ventriculoperitoneal shunt, neuronavigation

## Abstract

**Simple Summary:**

Hydrocephalus is the excessive accumulation of cerebrospinal fluid (CSF) in the ventricular brain. Surgical management is recommended for progressive neurological signs or unsuccessful medications. Ventriculoperitoneal shunt (VPS) is an option for improving clinical signs and long-term survival in dogs. This study compared the efficacy and postoperative overshunting between the low-pressure valve and medium-pressure valve VPS in dogs with severe hydrocephalus. The results showed overshunting in four dogs who received low-pressure valve placement, although there was still evidence of improved cognitive function and learning ability.

**Abstract:**

Hydrocephalus is a neurological disorder characterized by an abnormal accumulation of cerebrospinal fluid (CSF) within the ventricular system of the brain, leading to cerebral ventricular dilation, brain parenchyma compression, and neuronal cell loss. Surgery is an effective method of draining excessive amounts of CSF. Ventriculoperitoneal shunt (VPS) allows excess CSF to divert into the abdomen; this device is the most commonly used in the treatment of hydrocephalus both in veterinary and human patients. This study aims to describe the application of two types of VPS, low-pressure valve and medium-pressure valve, using a frameless stereotactic neuronavigational system in eight severe hydrocephalus in dogs and, in particular, analyze the prevalence of postoperative overshunting. Non-communicating hydrocephalus was found in seven dogs, whereas the rest of them had communicating hydrocephalus caused by traumatic brain injury with a skull fracture. The criteria for pressure valve selection depended on the intraoperative intraventricular pressure (IVP) that was determined by the adaptive manometer, according to the human protocol. Low-pressure valve placement was performed in five dogs, and the others received medium-pressure valve placement. The follow-up period was 2 weeks, 4–12 weeks, and 12 weeks to 12 months. Pre- and postoperative information including neurological signs, CT-Scan or MRI, medical treatment, complications, and ventricular volume were compared in all dogs. Seven dogs showed neurological improvement within 2 weeks after surgery. Overshunting was seen in four dogs who received low-pressure valve placement. Three of them had shunt infections within 4 to 6 weeks after surgery. One dog underwent shunt revision from a low-pressure valve to a medium-pressure valve caused by severe overshunting and progressive neurological signs. In addition, cognitive and learning improvements were evaluated based on the owners’ feedback, and neurological signs were examined during the follow-up period in two dogs that received low-pressure valve placement. We conclude that a medium-pressure valve is recommended for overshunting prevention. However, low-pressure valve placement seems to improve cognitive function and learning ability, which is related to an increase in the brain parenchyma observed during long-term monitoring. Moreover, we also report our experience and surgical procedure for frameless stereotactic ventricular shunt placement (FSVSP) in VPS surgery in dogs affected by hydrocephalus.

## 1. Introduction

Hydrocephalus is the excessive accumulation of cerebrospinal fluid (CSF) within the ventricular brain system [1,2,3]. Treatment options include medical management and surgical correction. Medical management aims to decrease CSF production using various medications, including glucocorticoids, carbonic-anhydrate, diuretics, and omeprazole [4], and surgical management is recommended for progressive neurological signs or unsuccessful medication. A ventriculoperitoneal shunt (VPS) has been reported to improve clinical signs and long-term survival in dogs [5,6,7]. The CSF flow control depends on the valve, which has a considerable differential opening pressure. Most manufacturers provide fixed pressure valves in four or five categories as follows: low-low (10–30 mmH_2_O), low (30–45 mmH_2_O), medium (85–105 mmH_2_O), and high-pressure (145–170 mmH_2_O) valves [2]. However, there are no well-defined indications for selecting shunts or their outcomes. Complications in dogs after VPS placement have been reported, including shunt infection, disconnection, overshunting, kinking, coiling, and seizure, which mainly occur 6 months after the operation [8]. Overshunting was regularly observed after VPS placement with low-pressure valves in dogs with severe hydrocephalus with thin cerebral parenchyma [9]. Because of the excessive drainage of CSF into the peritoneal cavity, a decrease in the size of the ventricle results in the stretching of bridging veins and tears, following bleeding in the subdural space [8,9,10]. The characteristics of overshunting signs include ventricular collapses, small/slit ventricles, and subdural effusion, such as subdural hemorrhage, subdural hematoma, and subdural hygroma. These signs can be detected via computed tomography (CT-Scan) or magnetic resonance imaging (MRI) after surgery [11,12,13]. Overshunting can affect the progression of neurological signs and is an important cause of death due to immediate intracranial pressure changes [7,9]. Previous studies recommended placing medium- or high-pressure valves to avoid overshunting in dogs with severe hydrocephalus [2,5,9]. However, a comparative efficacy between low- and medium-pressure valves in dogs resulted in fewer outcome data. This study aims to describe the application of two types of VPS, a low-pressure valve and a medium-pressure valve, using a frameless stereotactic neuronavigational system to avoid the disposition of the proximal ventricular catheter placement in VPS surgery for treating severe hydrocephalus in dogs.

## 2. Materials and Methods

### 2.1. Treatment

Eight non-medically responsive severely hydrocephalic dogs were prospectively studied from January 2019 to December 2021 at the Neurology Center of Kasetsart University Veterinary Teaching Hospital (KUVTH). The study was performed following the approval protocols for animal care and by using the scientific research data of Kasetsart Univerisity (approval ID# ACKU64-VET-079) and the owner’s informed consent. The diagnosis of hydrocephalus was diagnosed using CT-Scan (GE Optima CT660 CT Scanner) or MRI (Siemens MAGNETOM ESSENZA 1.5T) according to the stereotactic imaging protocol. For the frameless stereotactic ventricular shunt placement (FSVSP) (BrainLAB, Feldkirchen-Munich, Germany) technique, the surgeon planned the entry point, direction of the proximal ventricular catheter, and depth of ventricular catheter length using the navigation system (Figure 1) according to the BrainLAB manufacturer’s protocol. The patient-to-image registration procedure was divided into laser face contour and/or anatomical pair point methods. For the laser face contour method, the surgeon used a Z-touch that scanned around the nose, forehead, ear, scalp, periorbital, and surgical area. The sum of these points provides a face contour recognized by the navigation software and superimposed over the 3D virtual face reconstruction. Furthermore, the anatomical pair point method was performed by first touching the important anatomical landmarks (right lateral canthus, left lateral canthus, nasion, and nose) of dogs with a soft touch pointer. For the precision of registration calculated by the navigational software, the surface registration results should deviate by less than 2.5 mm. The pre- and postoperative ventricular volumes were calculated using the Smart Brush program (Figure 2A). This program is based on smart contour expansion with a click of the mouse or a touch of the finger. An adaptive manometer was used to monitor the intraventricular pressure, which was connected to the proximal ventricular by inserting it into the ventricle (Figure 2B) [14]. The CSF-flow control low-pressure valve (30–45 mm H_2_O) (2.21–3.31 mmHg) and medium-pressure valve (85–105 mm H_2_O) (6.25–7.72 mmHg) of Medtronic PS Medical^®^ CSF Shunt Kits (Medtronic, Minnesota, USA) were used in this study (Figure 3). In the FSVSP procedure, the surgeon inserted the ventricular catheter with a navigated disposable stylet to perforate the meninges (Figure 4A); the proximal catheter’s direction during insertion was monitored by an auto-pilot view of the navigation software (Figure 4B). After that, the surgeon connected the distal ventricular catheter to the adaptive manometer and placed it at the same level as the skull position, and the intraventricular pressure was recorded. Depending on the intraoperative intraventricular pressure related to the range of the opening pressure valve, a low- or medium-pressure valve was selected for each dog individually. After surgery, the dogs underwent an X-ray in the lateral and dorsoventral views to evaluate the VPS position. All surgical procedures were performed by only one surgeon with fifteen years of experience in small-animal neurosurgery and a degree of Diplomate from the Thai Board of Veterinary Surgery.

### 2.2. Follow-Up

After surgery, the dogs had follow-up and neurological examinations at 2 weeks, 4–12 weeks, and 12 weeks to 12 months. All owners were contacted to evaluate their feedback and the neurological signs they observed at the time of the study. A postoperative CT-Scan or MRI was performed at 12 weeks. Pre- and postoperative information included neurological signs, CT-Scan or MRI, medical treatment, complications, and ventricular volume, which were compared among all dogs. Other intraoperative data included patient-to-image registration method (anatomical pair point or laser face contour), animal preparation and registration times, VPS procedure time, and intraventricular pressure, which were also recorded. Cognitive and learning improvements were evaluated based on the information provided by the owners, including the interaction of the dogs with their owners and other pets, house soiling, and their response to training.

### 2.3. Statistical Analysis

This research involved a case study with a sample size of 5 dogs in the low-pressure valve group (LPVs) and 3 dogs in the medium-pressure valve group (MPVs). The patient data included the dog’s age, body weight, duration of signs, intraoperative intraventricular pressure, pre- and postoperative ventricular volume, animal preparation and registration times, and VPS procedure time, which were presented as mean ± SD. All information was summarized using descriptive statistics.

## 3. Results

### 3.1. Low-Pressure Valve Group (LPVs)

#### 3.1.1. Epidemiological Data

Information for LPVs (cases 1, 2, 3, 4, and 5) is summarized in Table 1. One dog (case 5) was an intact female. Others were intact males. The mean age was 6 ± 3.74 months (2–12 months). The mean body weight was 6.02 ± 2.53 kg (1.6–13.7 kg). The mean duration of neurological signs was 10.2 ± 8.49 weeks (3–24 weeks). Four of the five dogs (cases 1, 2, 3, and 5) were diagnosed via CT-Scan, and the other (case 4) was diagnosed via MRI. In all dogs, preoperative imaging (CT-Scan or MRI) revealed severe enlargement of the ventricles and thinning of the cerebral tissue. The fourth ventricle was unchanged in all dogs. The cause of hydrocephalus was classified as non-communicating that included mesencephalic aqueduct stenosis (cases 1, 3, and 4), mesencephalic aqueduct stenosis with arachnoid cyst (case 5), and obstruction of the interventricular foramen (Foramen of Monro) (case 2) [15]. The most common neurological signs in dogs were disorientation (cases 1, 4, and 5), obtundation (cases 2 and 3), aggressiveness (cases 1 and 5), circling (cases 1, 2, 4, and 5), right leaning (case 3), dysmetria (cases 2 and 4), seizure (case 5), periodic oral automatism signs (case 2), reduced or absent menace in both eyes (cases 1, 4, and 5), reduced or absent menace in only one eye (case 2), and positional conjugated nystagmus (case 1). In addition, five dogs had intact pupillary light reflexes (PLR) and normal ophthalmological examination. Pre- and postoperative neurological signs in LPVs are summarized in Table 2. Preoperative medical treatment for the management of increased intracranial pressure, including mannitol (0.25–1.0 g/kg, IV) and furosemide (0.7 mg/kg, IV), was used in three dogs (cases 2, 3, and 4). Other oral medications included the following: prednisolone (0.5 mg/kg, orally, once daily) was administered to one dog (case 1); dexamethasone (0.07–0.1 mg/kg, orally, once daily) was administered to four dogs (cases 2, 3, 4, and 5); furosemide (1 mg/kg, orally, once or twice daily) was administered to two dogs (cases 1 and 2); acetazolamide (10 mg/kg, orally, twice daily) was administered to two dogs (case 3 and 4); omeprazole (3 mg/kg, orally, twice daily) was administered to one dog (case 5). In addition, case 5 was administered phenobarbital (3 mg/kg, orally, every 12 h) as an anticonvulsant treatment for seizures, and case 2 was administered phenobarbital (2.5 mg/kg, PO, q12h) to control periodic oral automatism. The pre- and postoperative medical data for LPVs are summarized in Table 3.

#### 3.1.2. Intraoperative Data

The anatomical pair point method was used in three of the five dogs (cases 1, 2, and 3) for patient-to-image registration. For the other cases (cases 4 and 5), the laser face contour combined with the anatomical pair point method was performed for patient-to-image registration. The mean intraoperative IVP was 2.72 ± 0.61 mmHg (2.20–3.68 mmHg). All dogs had a unilateral VPS on the left side of the parietal bone. The mean preoperative ventricular volume was 134.14 ± 54.5 cm^3^ (52.2–205.4 cm^3^). The mean animal preparation and registration times were 87.8 ± 15.05 min (81–111 min). The mean VPS procedure time was 110.4 ± 28.96 min (87–159 min).

#### 3.1.3. Complication

Subdural effusion on postoperative imaging was revealed in four of the five dogs (cases 1, 2, 3, and 4); one dog (case 4) had severe ventricular dilation with a large open fontanelle area (Figure 5A). This case showed an acute progressive forebrain sign caused by overshunting within 72 h after surgery. The dog had severely depressed and horizontal nystagmus in the left-to-right direction. Physical examination revealed a collapse of the skin around the open fontanelle area and exophthalmos in the left eye (Figure 5B). Ultrasonography revealed bilateral subdural effusion (Figure 5C). Based on acute progressive signs, CSF was collected from the reservoir. The CSF analysis showed bleeding, and the CSF culture revealed no bacterial growth. Six days after surgery, the dog had a progressive right head turn, and stuporous and generalized seizures; these signs may involve intracranial hypertension. So, the medical management administered mannitol (0.5 g/kg, IV, every 24 h), furosemide (0.7 mg/kg, IV, every 24 h), and dexamethasone (0.1 mg/kg, IV, every 24 h) for three days; phenobarbital (3 mg/kg, PO, q12h) and levetiracetam (20 mg/kg, PO, q8h) were given as antiepileptic drugs. Later, the postoperative MRI at 12 weeks revealed severe bilateral hygroma with proximal catheter displacement into the subdural space (Figure 5D). So, this dog (case 4) received revision surgery to change from a low-pressure valve to a medium-pressure valve. The other three dogs (cases 1, 2, and 3) were not clinically overshunted; however, they had complications of infection within 4 to 6 weeks after surgery. In addition, these dogs showed progressive forebrain signs, including generalized seizures (cases 1 and 2), relapsing to circling (cases 1, 2, and 3), and abnormal mentation (cases 1, 2, and 3) with leukocytosis. However, none of the case studies included more CSF cytology or culture examination, but the flushing reservoir test identified normal shunt functions. A postoperative CT-Scan at 12 weeks revealed subdural hematoma with post-contrast enhancement around the cerebral cortex adjacent area in all dogs (Figure 6A–F), and one dog (case 3) had a small or slit ventricle (Figure 6F). However, all of them showed improvement in neurological signs after the medical management administered mannitol (0.5 g/kg, IV, every 24 h), furosemide (0.7 mg/kg, IV, every 24 h), and dexamethasone (0.07–0.1 mg/kg, IV, every 24 h) for three days during the subacute progression of neurological signs. A combination of antibiotics including enrofloxacin (7 mg/kg, orally, once daily) and cefixime (10 mg/kg, orally, twice daily) was prescribed for 4 to 6 weeks during the infection period. Another dog (case 3) experienced a postoperative seizure immediately after being given tramadol (4 mg/kg, IV) during the postoperative period of 24 h. Then, phenobarbital (2.5 mg/kg, PO, q12h) was started as an anti-epileptic drug. Nevertheless, case 3 had no seizures again during the follow-up period.

#### 3.1.4. Outcome

One dog (case 1) died because of a parvovirus infection, which was concurrent with status epilepticus. This dog had not been vaccinated before. Another dog (case 4) died because of progressive seizure to status epilepticus two months after receiving shunt revision. Four of the five dogs (cases 1, 2, 3, and 5) showed improvement in consciousness and gait within 2 weeks after surgery. Two of the five dogs (cases 2 and 3) showed marked improvement in mental alteration, learning ability, and house training for 4–12 weeks after surgery until the postoperative follow-up at 12 months. Case 5 had non-detected radiographic overshunting signs. The postoperative CT-Scan revealed a mildly reduced ventricular size with a ventricular volume of 46.1 cm^3^. In this case, repeated postoperative CT-Scan 6 months after surgery revealed that the ventricular size continued to reduce without overshunting signs; ventricular volume was 40.1 cm^3^ and the brain parenchyma had mild re-expansion (Figure 7A–C). At the twelve-month follow-up, the dog still had dullness, but was seizure-free and showed no aggressive behavior. Moreover, blindness in all the dogs was not resolved. The mean postoperative ventricular volume reduction percentage in yes, LPVs was 54.45 ± 29.4 (11.68–94.55%). Three dogs (cases 2, 3, and 5) continued medical treatments, which were slowly tapered off based on the clinical effects.

### 3.2. Medium-Pressure Valve Group (MPVs)

#### 3.2.1. Epidemiological Data

Information for MPVs (cases 6, 7, and 8) is summarized in Table 4. All the dogs were of the Chihuahua breed. The mean age was 21 ± 23.51 months (5–48 months). One dog (case 7) was a neutered female. Others were intact males. The mean body weight was 2.53 ± 1.15 kg (1.4–3.7 kg). The mean duration of neurological signs was 28.66 ± 21.19 weeks (6–32 weeks). Two of the three dogs (cases 7 and 8) were radiologically diagnosed via MRI, and another dog (case 6) was diagnosed via CT-Scan. The cause of hydrocephalus was classified as non-communicating (cases 6 and 7); e.g., obstruction of the lateral apertures with encephalitis (case 6), and obstruction of the lateral apertures with meningoencephalitis of unknown origin (case 7). Case 8 had communicating hydrocephalus caused by traumatic brain injury. The cytology analysis of CSF in case 6 revealed mononuclear pleocytosis; the total nucleated cell (TNCC) was 12 cells (RI: <5 cells/µL), and the protein level was 402 mg/dL (RI: <27 mg/dL). In case 8, CSF cytology revealed mild mononuclear pleocytosis with ventricular epithelial lining cells (TNCC 10 cells; RI: <5 cells/µL) and increased protein levels (33 mg/dL; RI: <27 mg/dL). All of the dogs were disoriented, unable to walk, and circled. Two dogs (cases 6 and 7) had unconjugated nystagmus. Other neurological signs were refractory seizures (cases 7 and 8), periodic oral automatism signs (case 6), and absence of menace in both eyes (cases 6, 7, and 8). In addition, two dogs (cases 6 and 8) showed an intact PLR and normal ophthalmological examination. Case 7 had subcortical blindness and was not responsive to the PLR test. Pre- and postoperative neurological signs in MPVs are summarized in Table 5. Preoperative medical treatment, including mannitol (0.5 g/kg, IV) and furosemide (0.7 mg/kg, IV), was administered in one dog (case 6). Other oral medications included the following: prednisolone (1 mg/kg, orally, once daily), was administered to one dog (case 7); dexamethasone (0.05–0.1 mg/kg, orally, once daily) was administered to two dogs (cases 6 and 8); furosemide (0.7 mg/kg, orally, once daily) was administered to one dog (case 6); acetazolamide (5–10 mg/kg, orally, twice daily) was administered to two dogs (case 7 and 8). In addition, two dogs (cases 7 and 8) were administered phenobarbital (3.5 mg/kg, orally, every 12 h) as an anticonvulsant treatment for seizures, and one dog (case 6) was administered phenobarbital (3 mg/kg, orally, every 12 h) for oral automatism. Two dogs (cases 6 and 8) were administered pregabalin (5 mg/kg, orally, every 8 h) for sedation. Levetiracetam (26–30 mg/kg, orally, every 8 h) was combined with phenobarbital as an anticonvulsant in two dogs (cases 7 and 8). The pre- and postoperative medical treatment information for MPVs is summarized in Table 6.

#### 3.2.2. Intraoperative Data

The anatomical pair point method was used in three dogs (cases 6, 7, and 8) for patient-to-image registration. The mean IVP was 7.34 ± 1.46 mmHg (7.35–8.8 mmHg). Two dogs (cases 7 and 8) had a unilateral VPS on the left side of the parietal bone, and the other dog (case 6) had a unilateral VPS on the left side and proximal catheter insertion through the persistent open fontanelle. The mean preoperative ventricular volume was 25.7 ± 20.95 cm^3^ (11–49.7 cm^3^). The mean animal preparation and registration times was 121.66 ± 45.65 min (74–165 min). The mean VPS procedure time was 102.33 ± 20.42 min (79–117 min).

#### 3.2.3. Complication

None of the dogs had serious postoperative complications. Nonetheless, case 6 showed signs of recurrent worsening after reducing the doses of steroids and diuretics. These results were related to the mild attachment of the proximal ventricular catheter to the brain parenchyma (Figure 8A,B) or concurrent high CSF protein that induced inappropriate drainage. However, the pump flushing test was normal.

#### 3.2.4. Outcome

All of the dogs in MPVs showed improvement in behavioral performance. Two dogs (cases 6 and 7) showed reduced vocalization, and case 8 showed reduced aggressive behavior 2 weeks after VPS. In addition, two dogs (cases 6 and 8) showed improvement in gait; they could stand and walk for 2 weeks after surgery. However, case 7 attempted to stand and right circled; these signs continued to improve for more than 12 weeks after surgery. This dog was seizure-free until 12 months in postoperative monitoring. Case 8 had decreased seizure frequency for 2 weeks after surgery. However, this dog was not followed-up at 4–12 weeks after surgery, and postoperative imaging at 12 weeks was not performed. Subsequently, 6 months after surgery, case 8 had a progressive right head turn, seizure, and inability to walk. However, the dog showed no response to medical treatment, and the owner decided on palliative treatment for the dog. The percentage of postoperative ventricular volume reduction in two dogs (cases 6 and 7) was 13.53 ± 6.67% (8.85 to 18.29%). Postoperative CT-Scan in cases 6 and 7 showed a mildly decreased ventricular size, mild brain parenchyma re-expansion, and non-detectable overshunting. None of the dogs showed improved cognitive function or learning ability in long-term monitoring for 12 months after surgery.

## 4. Discussion

Our inclusion criteria for case selection were based on neurological assessments and preoperative CT-Scan or MRI findings, which revealed severe enlargement of the ventricle. All dogs were initially treated medically and either failed to improve or relapsed during the treatment. This study used the CSF-flow control valve types, which were fixed pressure valves with a reservoir dome that collected CSF samples and tested shunt function. The pumping test benefited from the differentiation and interpretation of partial or total obstruction of the ventricular catheter system [16]. Our IVP monitoring technique was manipulated using an adaptive human manometer. The manometer was used in the lumbar puncture to obtain indirect intracranial pressure measurements in humans [14,17]. For our results, the IVP of dogs in low-pressure valve placement (cases 1, 2, 3, 4, and 5) was 2.72 ± 0.61 mmHg (2.2–3.68 mmHg), and the IVP of dogs in medium-pressure valve placement (cases 6, 7, and 8) was 7.34 ± 1.46 mmHg (7.35–8.8 mmHg). These results are in agreement with a previous study where the mean IVP in dogs with communicated hydrocephalus was 8.8 ± 4.22 mmHg (3–18 mmHg) [18]. The finding of normal or low IVP in most dogs with hydrocephalus could also depend on the time point of measurement. It is also possible that an initially high IVP returns to the typical values in the late stage of the disease. In our study, we evaluated individual intraoperative IVP and opening pressure (low-pressure valve (30–45 mm H_2_O) (2.21–3.31 mmHg) and medium-pressure valve (85–105 mm H_2_O) (6.25–7.72 mmHg)) for valve pressure selection. As the valve regulates normal intraventricular pressure and the pressure is high in the opening pressure threshold, the valve opens to allow CSF drainage to the peritoneal cavity. When the pressure falls below the closing pressure threshold, the valve closes, thereby stopping the flow of CSF [19]. If the valve-opening pressure does not match the cerebrospinal fluid (CSF) hydrodynamics of the patient, CSF overshunting or undershunting signs can develop. For this reason, the adjustable valve is used to restrict inappropriate CSF shunting, which allows the clinician to adjust the opening pressure after the shunt is implanted using a device that emits a magnetic field [20]. Nevertheless, an adjustable shunt valve is not used worldwide in animals because of financial problems. Further, the anti-siphon effect is also used to prevent overshunting, which occurs during the patient changes position, especially in upright times. In dogs, the siphoning outcome is at lower risk because the shunt lining is more horizontal, with the head roughly equal to the abdomen.

Most neurological signs in our study included forebrain dysfunction, especially blindness with intact PLR, gait deficits, and seizures. However, other signs showed improvement after VPS surgery, but all cases remained blind in long-term monitoring. Persistent central blindness is probably a consequence of structural damage to the visual pathways in the periventricular white matter and compression of the lateral geniculate body [21,22]. Abnormal gait and movement disorders in affected animals can vary depending on the obstruction level, which indicates a block of CSF flow in the cerebellum or brain stem. For abnormal gait, six dogs (cases 1, 2, 4, 6, 7, and 8) had circling. As for the others, cases 2 and 4 exhibited dysmetria gait caused by severe CSF accumulation that affected cerebellar compression. Cases 6 and 7 had unconjugated nystagmus, which might be related to a massive dilation of the fourth ventricle, including impairment of CSF flows through the lateral apertures. Nonetheless, almost all dogs showed an improvement in gait after VPS placement within 2 weeks after surgery. These results may be associated with the insertion of the ventricular catheter and manual CSF removal during the procedure. In case 6, the result showed recurrent worsening signs after the reduction in the dose of diuretics and steroids. These results may be related to improper draining because of the attachment of the proximal catheter to the brain parenchyma and the high CSF protein concentration (402 mg/dL). In general, blood and protein are other causes of shunt catheter occlusion in humans. However, the CSF protein level in dogs and cats is lower than that in humans (<30 mg/dL compared to 50–150 mg/dL); therefore, the obstruction of the proximal catheter by protein occlusion has a lower incidence in dogs [8]. Two cases (cases 2 and 6) showed preoperative periodic oral automatism, in which case 6 had been detected with severe vocalization. These clinical signs may indicate a focal epileptic seizure, which is elicited by cortical stimulation. These neurological signs have been described in the mesial temporal lobe, mesial frontal lobe, and cingulate gyrus in humans [23]. However, one week after phenobarbital administration, both dogs showed a decrease in the frequency of oral automatism; subsequently, it disappeared in case 2 after 12 weeks of VPS placement. These signs can be further investigated via electroencephalogram (EEG) for differentiation from focal seizures. In addition, refractory seizures were seen in cases 7 and 8, who had coincidental hydrocephalus with other causes, such as inflammatory brain disease and traumatic brain injury. Usually, seizures are observed less than the other neurological signs in patients with hydrocephalus. In a previous study, seizure prevalence in dogs and cats diagnosed with internal hydrocephalus was common and non-observed for two years after VPS [24]. These results were related to the experimental dog and cat models; the seizures did not occur after inducing ventricular dilation to four-fold of the average level [25,26]. Postoperative seizures were seen in case 3 on the day after VPS placement. This dog had an acute generalized tonic-clonic seizure after being given intravenous tramadol (4 mg/kg). The seizure event, in this case, may have occurred because of the side effects of tramadol, which inhibits noradrenaline reuptake that involves acute seizure, as in a human report [27]. In the other cases, seizure events were concurrently noticed with infection in cases 1 and case 2 and were acquired after acute overshunting in case 4. Three dogs (cases 1, 2, and 3) showed progressive neurological signs with leukocytosis during postoperative weeks 4–6, who were suspected to have shunt infection. Unfortunately, none of them had been investigated further during the infection time, including CSF cytology or culture examination, but the flushing reservoir test identified normal functions of VPS. Nevertheless, a postoperative CT-Scan at 12 weeks in these dogs revealed a subdural hematoma with post-contrast enhancement around the cerebral cortex adjacent area, which was characteristic of empyema signs, similar to that in humans [28]. Shunt infection occurred in 8.5% of retrospective multicenter case series of dogs and cats [7]. Some of the common bacteria are *Staphylococcus* spp., especially *Staphylococcus epidermis* [29]. In this study, we used ceftriaxone and enrofloxacin for bacterial coverage in these cases. All of them showed improved neurological signs after medical management. Currently, antibiotics used for the prevention of shunt infections can be administered in five ways: orally; intravenously; intrathecally; topically; and via the implantation of antibiotic-impregnated shunt catheters [30]. Previously, antibiotics given via the oral route were used as add-on therapy in the treatment of CSF-shunt infections but are rarely used to prevent CSF-shunt infections [31]. A new technique in the field of shunt infection prevention is the antibiotic-impregnated catheter. These catheters, which are impregnated with two antibiotic agents (vancomycin and gentamicin), slowly release antibiotics over a period of days, and they have been shown to significantly reduce the rate of shunt infections [32]. In one study, the patient group that received an injection of prophylactic vancomycin and gentamicin in the reservoir and around the peritoneal catheter showed a significant reduction in the incidence of postoperative shunt infection [33].

In some previous reports, excessive shunting and associated ventricular collapse with subdural hematomas or subarachnoid hemorrhages are potential postoperative complications found in 2.7–2.8% of dogs [7,8]. In our study, subdural effusion on postoperative radiology was revealed in four of the five dogs (cases 1, 2, 3, and 4). The cause of acute clinical overshunting in case 4 may have emerged from the large open fontanelle area, which induced more ventricular collapse than in other cases (cases 1,2, and 3). Similar complications were reported in a previous case study [9]. Three cases (cases 1, 2, and 3) had chronic overshunting signs, which were diagnosed via postoperative CT-Scan. Unfortunately, all of the dogs had concurrent shunt infections. These dogs responded to acute worsening neurological signs during mannitol infection, which may be associated with intracranial pressure reduction. However, these dogs lacked immediate radiological monitoring; therefore, subacute overshunting could not be ruled out. In the comparison between low- and medium-pressure valve placements, overshunting was observed in dogs (cases 1, 2, 3, and 4) that received low-pressure valve placement, but it was not detected in all the dogs that received medium-pressure valve placement (cases 6 and 7). Cases 2 and 3 showed an improvement in cognitive function that correlated with the re-expansion of the cerebral brain parenchyma. Later, steroid and diuretic dosages were reduced in both dogs without deterioration. In contrast, postoperative imaging in cases 6 and 7 showed a mild decrease in ventricular size and a slightly increased brain parenchyma. Interestingly, these dogs showed improvements in seizure and gait, but cognitive function did not resolve after postoperative follow-up at 12 months. For these results, the improvement in cognitive function and learning ability may be related to an increase in brain parenchyma after VPS placement. In a previous study, cognitive dysfunction could be resolved after VPS placement, depending on the resolution of periventricular white matter edema alone [19]. However, our study had a limitation on the radiological efficiency for precise diagnosis because some dogs were not investigated postoperatively with MRI. In our study, postoperative imaging was performed at 12 weeks, which may not be a long enough time. Therefore, we suggest repeating the postoperative imaging 6 months after surgery or when recurrent neurological signs are detected. Repeated postoperative CT-Scan in case 5 revealed a continued decrease in the ventricular volume and mild re-expansion of the brain parenchyma without overshunting signs. Based on these results, we hypothesize that case 5 received VPS surgery after the clinical signs started 24 weeks before surgery. So, prolonged compression of the brain parenchyma may lead to neuronal tissue degeneration and dysfunction. However, this dog still had abnormal mentation for a long time, as monitored during the follow-up period.

The frameless stereotactic neuronavigational system has been used in several neurosurgical procedures in animals for brain biopsy and brain tumor excision [34,35]. However, this system lacks periodic reports on VPS placement in animals. In a previous study, the intraoperative navigation system for VPS improved patient outcomes and the accuracy of ventricular catheter placement more than conventional surgery in humans [36]; therefore, the neuronavigational system is a precise technique for ventricular catheter placement. Fortunately, most of the dogs in this study did not demonstrate complications with proximal catheter occlusion, and postoperative imaging revealed suitable proximal catheters in the ventricle.

For our FSVSP, the patient-to-image registration procedure was performed using the laser face contour with Z-touch and/or anatomical pair point techniques; these are satisfactory and preserve time more than the fiducial skin marker. In a previous study, there was no statistical difference in the registration techniques between the standard fiducial and anatomical landmarks and the optical neuronavigational-guided intracranial biopsy procedures in horses [35]. Five of the eight dogs (cases 1, 2, 3, 6, 7, and 8) underwent registration from the anatomical pair point. Two dogs (cases 4 and 5) were registered by laser face contour and the registration was verified by anatomical pair point. The mean animal preparation and registration times in all dogs was 100.5 ± 32.12 min (70–165 min). The problem encountered in our FSVSP procedure is the prolonged registration time. Frequently, the Z-touch laser cloud did not detect the skin surface. This is because dogs have more hair covering their skin than humans do. This may have affected the surface matching deviation by more than 2.5 mm. We suggest that the dogs should have their head fixed with a headstand, after which the anatomy landmarks following the right lateral canthus, left lateral canthus, nasion, and nose should be evident because these locations are significant to the accuracy of the patient registration procedure according to the manufacturer’s protocol. Another anatomical landmark area should be to thoroughly clean the hair. In this study, the Smart Brush program was used for volume calculation to compare pre- and postoperative ventricular sizes. This technique has been reported to monitor intracranial arachnoid cyst volume in children after cysto-peritoneal shunt surgery [37]. Generally, morphological criteria include the ventricle/brain (VB)-index, which is a two-dimensional volumetric measurement used for radiological evaluation in dogs with hydrocephalus [38]. In our opinion, this software offers a greater consistency in volume measurement and is suitable for use in clinical practice to compare postoperative VPS outcomes. In this study, the neuronavigational system was a precise technique for ventricular catheter placement. However, the FSVSP procedure requires proficiency and takes more time to equip longer than conventional VPS.

## 5. Conclusions

Based on our results, the low-pressure valve can cause overshunting more frequently than the medium-pressure valve. However, the low-pressure valve may be more suitable for increasing brain parenchyma volume and improving learning ability than the medium-pressure valve. Nevertheless, this study was limited by its small number of cases. Furthermore, the correlations between low- and medium-pressure valves with learning improvement should be considered in future studies.

## Figures and Tables

**Figure 1 animals-13-01890-f001:**
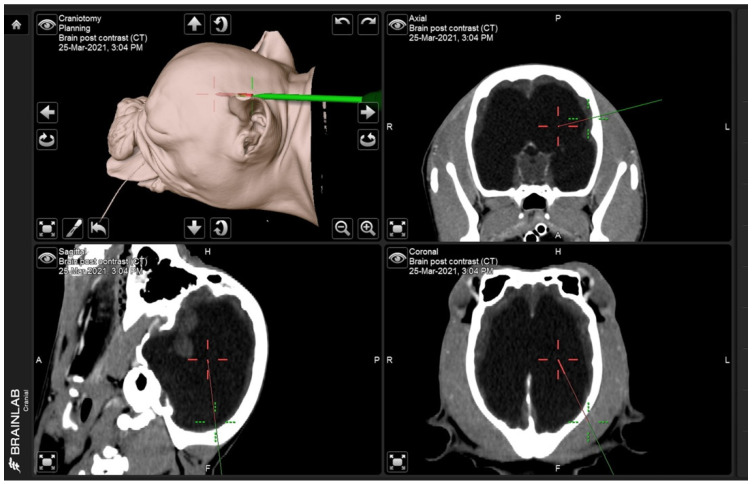
Screen capture shows the entry point (green line) and the target direction of the proximal catheter (red line) from three angles: sagittal, transverse, and coronal.

**Figure 2 animals-13-01890-f002:**
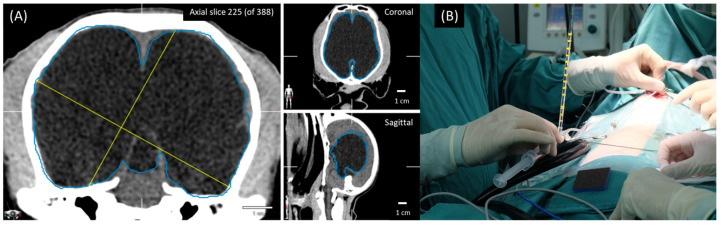
Ventricular volume calculated by Smart Brush program (**A**). The adaptive manometer was connected to the distal end of the proximal ventricular catheter after ventricle insertion (**B**).

**Figure 3 animals-13-01890-f003:**
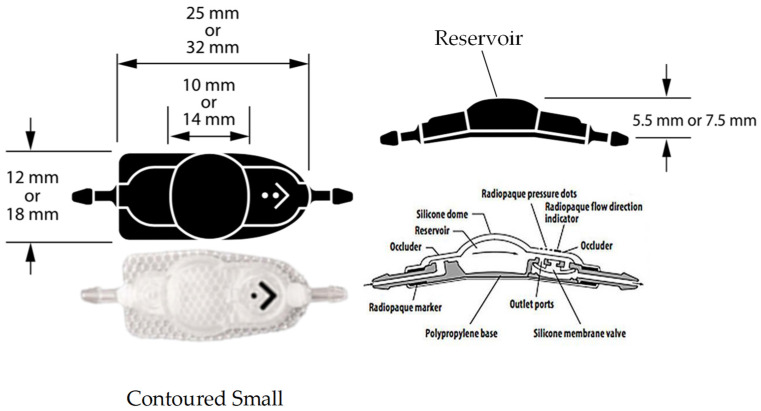
The CSF-flow control valve of Medtronic PS Medical^®^ CSF Shunt Kits, which were fixed pressure valves with a reservoir dome.

**Figure 4 animals-13-01890-f004:**
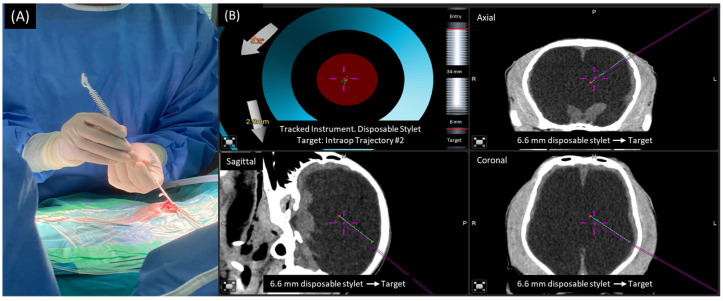
The navigated disposal stylet was used as a navigation tracker while the surgeon inserted the proximal catheter into the ventricle (**A**). Screen capture shows the target direction of the proximal catheter from three angles (green line) and direction of navigated disposal stylet (purple line) from three angles,, which the surgeon used for intraoperative monitoring (**B**).

**Figure 5 animals-13-01890-f005:**
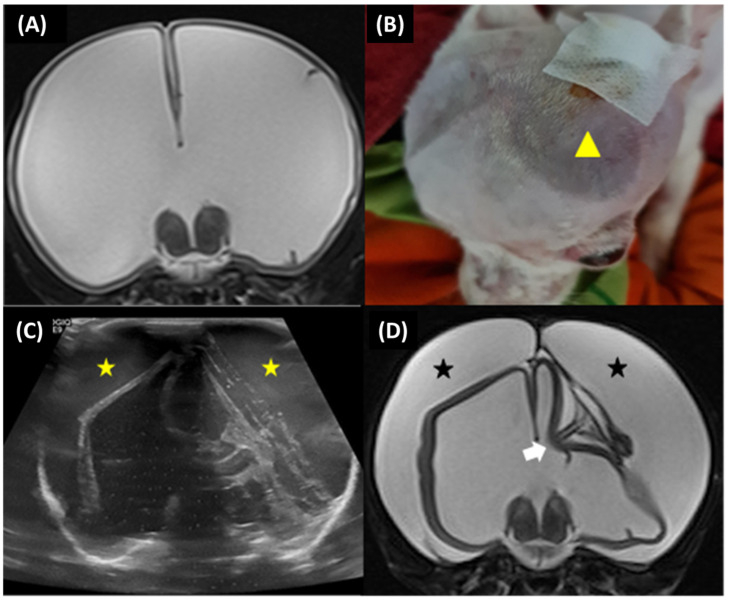
Case 4, 6-month-old male Jack Russel Terrier. Preoperative transverse T2-weighted MRI revealed severe ventricular dilatation with thin brain parenchyma (**A**). The dog had collapsed skin around the open fontanelle area (yellow arrowhead) (**B**). Postoperative ultrasonography at 72 h revealed bilateral subdural effusion (yellow star) (**C**). Postoperative transverse T2-weighted MRI at 12 weeks revealed severe bilateral subdural hygroma (black star) with the proximal catheter in an incorrect position (white arrow) (**D**).

**Figure 6 animals-13-01890-f006:**
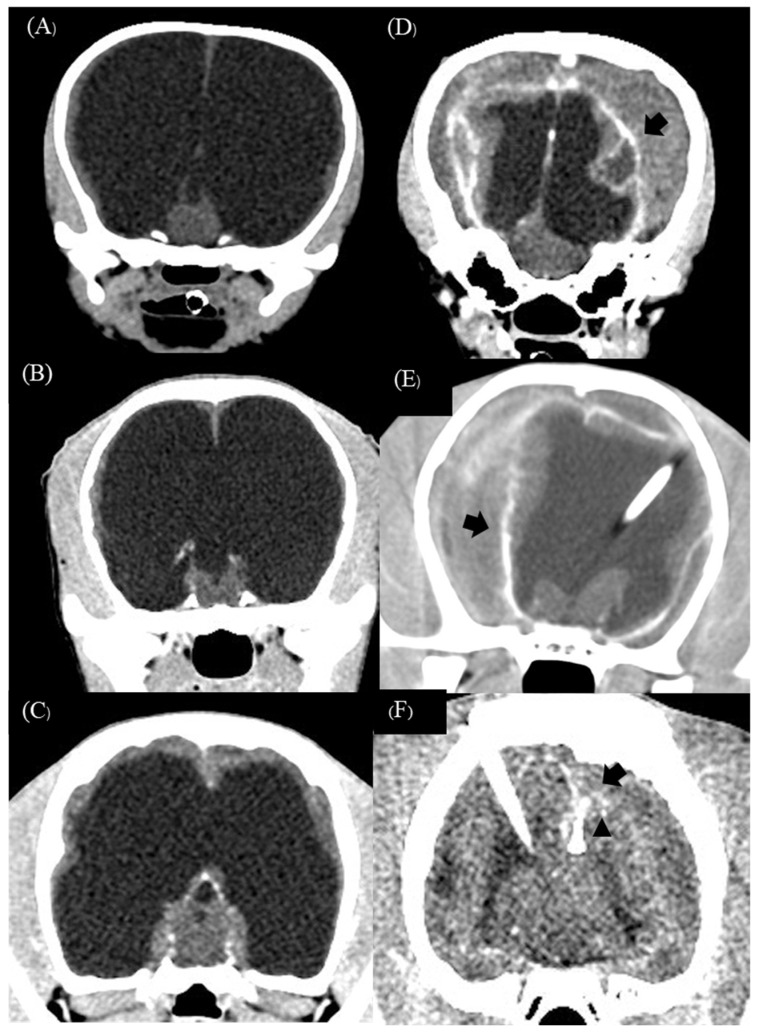
Preoperative transverse CT-Scan of case 1 (**A**), case 2 (**B**), and case 3 (**C**) revealed severe ventricular enlargement with thin brain parenchyma. Postoperative transverse CT-Scan post-contrast at 12 weeks revealed bilateral effusion with adjacent cortex contrast enhancement (sign of empyema) (black arrow) in three dogs (**D**–**F**). Postoperative transverse CT-Scan revealed a small ‘slit’ ventricle (black arrowhead) in case 3 (**F**).

**Figure 7 animals-13-01890-f007:**
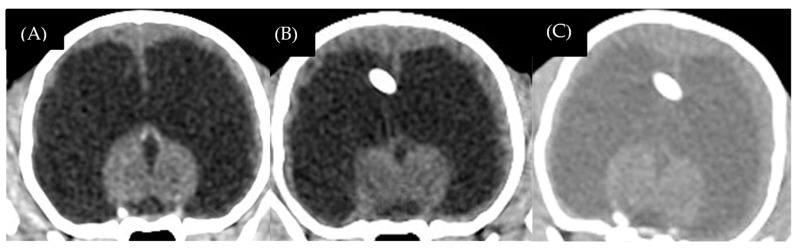
Case 5, 3-month-old male Chihuahua. Preoperative sagittal CT-Scan revealed ventricular dilation with thin brain parenchyma (**A**). Postoperative transverse CT-Scan at 12 weeks revealed a mild decrease in ventricular size (**B**). Postoperative transverse CT-Scan at 6 months revealed mild brain parenchyma re-expansion (**C**).

**Figure 8 animals-13-01890-f008:**
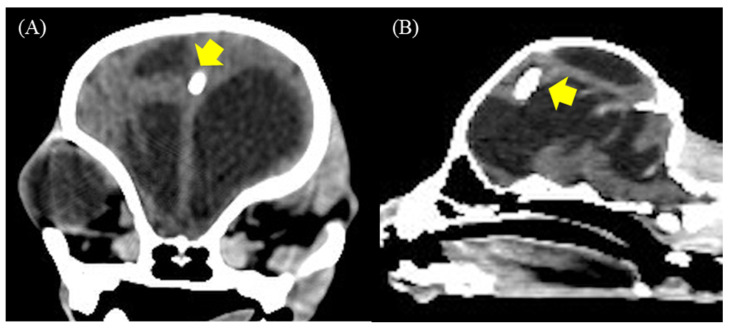
(**A**,**B**) Postoperative transverse and sagittal CT-Scan at 12 weeks revealed a proximal ventricular catheter mildly attached to the brain parenchyma (yellow arrow) in case 6.

**Table 1 animals-13-01890-t001:** Epidemiological data and results of intraventricular pressure, pre- and postoperative ventricular volume, and postoperative imaging in LPVs.

Low-Pressure Valves (LPVs)											
Case	Breed	Sex	Age (Month)	Body Weight (kg)	Duration of Signs (Week)	Cause of Hydrocephalus	Intraventricular Pressure (mmHg)	Pre-Ventricular Volume (cm^3^)	Post-Ventricular Volume 12 Weeks (cm^3^)	Ventricular Volume Reduction (%)	Post-Imaging Overshunting (Subdural Effusion)
1	Crossbreed	male	2	2.4	3	Mesencephalic aqueduct stenosis	2.94	142.10	60	57.77	Bilateral subdural hematoma and empyema
2	American Pitbull	male	4	13.7	8	Obstruction of interventricular foramen	2.57	205.40	98.60	51.99	Bilateral subdural hematoma and empyema
3	American Bully	male	6	9.7	4	Mesencephalic aqueduct stenosis	3.68	139	7.57	94.55	Bilateral subdural hematoma and empyema
4	Jack Russel Terrier	male	6	1.6	12	Mesencephalic aqueduct stenosis	2.20	132	57.70	56.28	Bilateral subdural hematoma
5	Chihuahua	female	12	2.7	24	Mesencephalic aqueduct stenosis with arachnoid cyst	2.21	52.20	46.10	11.68	Not detectable

**Table 2 animals-13-01890-t002:** Preoperative and postoperative neurological signs at 2 weeks, 4–12 weeks, and 12 months in LPVs.

Case	Preoperative Neurological Signs	Postoperative Signs (2 Weeks)	Postoperative Signs (4–12 Weeks)	Postoperative Signs (12 Months)
1	Disoriented Aggressive Right circling Conjugated nystagmus Both eye cortical blindness	Consciousness Reduced right circling Reduced aggressiveness	Shunt infection (6 weeks) Seizure Consciousness Reduced right circling Reduced aggressiveness	Death (parvovirus infection)
2	Obtunded Right circling Dysmetria Left eye cortical blindness Periodic oral automatism	Consciousness Reduced right circling Reduced dysmetria	Shunt infection (4 weeks) Seizure Consciousness Moderated right circling Improved learning ability Periodic oral automatism signs disappeared	Consciousness Continuously improved learning Mild right circling Seizure free Left eye cortical blindness
3	Obtunded Right leaning	Seizure (tramadol effect) Consciousness	Shunt infection (6 weeks) Consciousness Improved learning ability	Consciousness Continuously improved learning Seizure free
4	Disoriented Right circling Dysmetria Both eye cortical blindness	Disoriented Seizure Acute overshunting (72 h)	Disoriented Right circling and dysmetria Both eye cortical blindness Second shunt revision	Death (status epilepsy)
5	Disoriented Aggressive Both eye cortical blindness Seizure	Disoriented Both eye cortical blindness Decreased seizure frequency Reduced aggressiveness	Mild improvement in consciousness Both eye cortical blindness Decreased seizure frequency Reduced aggressiveness	Both eye cortical blindness Decreased seizure frequency No aggressiveness

**Table 3 animals-13-01890-t003:** Preoperative medical treatment and postoperative medical treatment at VPS 12 weeks and 12 months in LPVs.

Low-Pressure Valves (LPVs)			
Case	Preoperative Medical Treatment	Postoperative Medical Treatment at VPS 12 Weeks	Postoperative Medical Treatment at VPS 12 Months
1	Prednisolone 0.5 mg/kg, PO, SID Furosemide 2 mg/kg, PO, BID Gabapentin 10 mg/kg, PO, BID	Prednisolone 0.5 mg/kg, PO, SID Furosemide 1 mg/kg, PO, SID Phenobarbital 2.7 mg/kg, PO, q12h	-
2	Mannitol 0.5 g/kg, IV, q24h with furosemide 0.7 mg/kg, IV, q24h for 3 days Dexamethasone 0.1 mg/kg, PO, SID Phenobarbital 2.5 mg/kg, PO, q12h	Dexamethasone 0.02 mg/kg, PO, QOD Furosemide 1.5 mg/kg, PO, SID Phenobarbital 2.5 mg/kg, PO, q12h	Dexamethasone 0.02 mg/kg, PO, Q2D Phenobarbital 2 mg/kg, PO, q12h
3	Mannitol 0.5 mg/kg, IV, q24h Furosemide 0.7 mg/kg, IV, q24h Dexamethasone 0.1 mg/kg, PO, SID Acetazolamide 10 mg/kg, PO, BID	Dexamethasone 0.05 mg/kg, PO, SID Furosemide 1 mg/kg, PO, SID Phenobarbital 2.7 mg/kg, PO, q12h	Dexamethasone 0.02 mg/kg, PO, Q3D
4	Dexamethasone 0.1 mg/kg, PO, SID Acetazolamide 10 mg/kg, PO, BID	Dexamethasone 0.1 mg/kg, PO, SID Phenobarbital 3 mg/kg, PO, q12h Levetiracetam 25 mg/kg, PO, q8h	-
5	Phenobarbital 3 mg/kg, PO, q12h Dexamethasone 0.07 mg/kg, PO, SID Furosemide 1 mg/kg, PO, SID Omeprazole 3 mg/kg, PO, SID	Phenobarbital 3 mg/kg, PO, q12h	Phenobarbital 3 mg/kg, PO, q12h

**Table 4 animals-13-01890-t004:** Epidemiological data and results of intraventricular pressure, pre- and postoperative ventricular volume, and postoperative imaging in MPVs.

Medium-Pressure Valves (MPVs)											
Case	Breed	Sex	Age (Month)	Body Weight (kg)	Duration of Signs (Week)	Cause of Hydrocephalus	Intraventricular Pressure (mmHg)	Pre-Ventricular Volume (cm^3^)	Post-Ventricular Volume 12 Weeks (cm^3^)	Ventricular Volume Reduction (%)	Post-Imaging Overshunting (Subdural Effusion)
6	Chihuahua	male	5	1.4	6	Obstruction of the lateral apertures with encephalitis	8.80	49.70	45.30	8.85	No detectable
7	Chihuahua	female	48	2.5	48	Obstruction of the lateral apertures with meningoencephalitis of unknown origin	5.88	16.40	13.40	18.29	No detectable
8	Chihuahua	male	10	3.7	32	Post-traumatic	7.35	11	-	-	-

**Table 5 animals-13-01890-t005:** Preoperative and postoperative neurological signs at 2 weeks, 4–12 weeks, and 12 months in MPVs.

Case	Preoperative Neurological Signs	Postoperative Signs (2 Weeks)	Postoperative Signs (4–12 Weeks)	Postoperative Signs (12 Months)
6	Disoriented Vocalization Periodic oral automatism signs Unable to walk and turned to the left Unconjugated nystagmus Both eye cortical blindness	Disoriented Reduced vocalization Reduced periodic oral automatism Reduced nystagmus Left circling	Disoriented Reduced left circling Reduced nystagmus and periodic oral automatism	Disoriented Still left circling Still nystagmus and periodic oral automatism
7	Disoriented Severe vocalization Unable to walk and turned to the right Unconjugated nystagmus Refractory seizure Both eye subcortical blindness	Reduced vocalization, attempted to stand, and right circling Reduced nystagmus Seizure free	Attempted to stand and right circling No nystagmus Seizure free	Slight walking and no circling No nystagmus Seizure free Both eye subcortical blindness
8	Disoriented Aggressive Unable to walk and turned to the right Refractory seizure Both eye cortical blindness	Disoriented Reduced aggressiveness Right circling Decreased seizure frequency (2 weeks)	No further follow-up	Disoriented Unable to walk and turned to the right Refractory seizure

**Table 6 animals-13-01890-t006:** Preoperative medical treatment and postoperative medical treatment at VPS 12 weeks and 12 months in MPVs.

Medium-Pressure Valves (MPVs)			
Case	Preoperative Medical Treatment	Postoperative Medical Treatment at VPS 12 Weeks	Postoperative Medical Treatment at VPS 12 Months
6	Mannitol 0.5 mg/kg, IV, q24h Furosemide 0.7 mg/kg, IV, q24h Dexamethasone 0.1 mg/kg, PO, SID Phenobarbital 3 mg/kg, PO, q12h Pregabalin 5 mg/kg, PO, q8h	Phenobarbital 3 mg/kg, PO, q12h Dexamethasone 0.1 mg/kg, PO, QOD Furosemide 0.7 mg/kg, PO, QOD	Phenobarbital 3 mg/kg/q12h Dexamethasone 0.1 mg/kg/Q2D Furosemide 0.7 mg/kg/QOD
7	Phenobarbital 3.5 mg/kg, PO, q12h Levetiracetam 26 mg/kg, PO, q8h Pregabalin 5 mg/kg, PO, q8h Acetazolamide 5 mg/kg, PO, BID Prednisolone 1 mg/kg, PO, QOD	Phenobarbital 3.5 mg/kg, PO, q12h Levetiracetam 26mg/kg, PO, q8h Pregabalin 5 mg/kg, PO, q8h Prednisolone 0.25 mg/kg, PO, QOD	Phenobarbital 3.5 mg/kg, PO, q12h Levetiracetam 26 mg/kg, PO, q8h Pregabalin 5 mg/kg, PO, q8h Prednisolone 0.25 mg/kg, PO, QOD
8	Dexamethasone 0.05 mg/kg, PO, SID Acetazolamide 10 mg/kg, PO, BID Phenobarbital 3.5 mg/kg, PO, q8h Levetiracetam 30 mg/kg, PO, q8h	Dexamethasone 0.05 mg/kg, PO, SID Phenobarbital 3.5 mg/kg, PO, q8h Levetiracetam 30 mg/kg, PO, q8h (2 weeks after surgery)	Dexamethasone 0.05 mg/kg, PO, QOD Furosemide 1 mg/kg, PO, SID Phenobarbital 3.5 mg/kg, PO, q8h Levetiracetam 30 mg/kg, PO, q8h

## Data Availability

The data presented in this study are available upon request from the corresponding authors.

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
