# Peer review of "Evaluation of Overshunting between Low and Medium Pressure Ventriculoperitoneal Shunts in Dogs with Severe Hydrocephalus Using Frameless Stereotactic Ventricular Shunt Placement"

_animals, 2023, doi:10.3390/ani13121890_

Round 1

Reviewer 1 Report

Comment to the authors

Dear Authors, 

This is a very interesting study about the prevalence of overshunting between low and medium-pressure valve for severe hydrocephalus in dogs, but some comments need to be stressed out.

Overall, this manuscript provides a good description of the surgical procedure and follow up, analyses many different parameters, but in my opinion in some parts it is missing a proper revision for the quality of writing.

Please see some comments below:

Simple summary

Line 14_change “of” with “for” if the meaning of the sentence is maintained

Line 15_change “efficiency” with “efficacy”

Line 16_change in “…VPS in dogs with severe hydrocephalus.”

Line17_ modified as “The results showed overshunting in 4 dogs in which the low pressure valve was placed, although there was still evidence of improved learning.”

ABSTRACT

Line 22-23_ modify the sentence as follows ”Ventriculoperitoneal shunt (VPS) allows to divert the excess CSF into the abdomen: this device is the most commonly used in the treatment of hydrocephalus both in veterinary and in human patients.”

Line 24_ see if the meaning of the phrase is maintained and then modify as suggested: “This study aim to describe the application of two type of VPS, low-pressure valve and medium-pressure valve, using a frameless stereotactic neuronavigational system in 8 severe hydrocephalus dogs, in particular analyzing the prevalence of postoperative overshunting.”

Line 34_ modify with “…7 dogs had showed…”

Line 36_ modify with “One dog underwent shunt revision…”

Line 43_ modify with “ …in dogs affected by hydrocephalus”

INTRODUCTION

Line 55_“The CSF flow control depends on the valve, which has a differential opening pressure.”

Line 60_ what do you mean with “kinging”? Is a type error for the “kindling phenomenon” or for the “kinking” of the catheter? Please modify/correct.

Line 74-77_ please, see the same correction made on the abstract.

MATERIALS AND METHODS

Line 83-84_what do you mean with “definitive diagnostic”? That the diagnosis of Hydrocephalus was made with CT? Please modify this with a more clear sentence.

RESULTS

Line 114_ “preoperative radiography” you mean CT scan? Please modify. See also later in the text line 93 and 07 of the Discussion.

Line 126_ there is a red point after the word “Table 2”.

CONCLUSION

Line 147-148_ please, modify as  “However, the low-pressure valve may be suitable for increasing brain parenchyma volume and to improving learning ability more than medium pressure one.”

Line 149-151_  please, see if the meaning of the phrase is maintained and then modify as suggested “Nevertheless, this study had a limitation of the small number of cases included. Further, the correlations between low- and medium-pressure valves with learning improvement will be considered in future studies.”

Did you utilized a rating scale to demonstrate that the dogs treated improved or not in learning? Or is just a feedback from the owners? Did you administered a sort of questionnaire for the owner to monitored the dogs’ brain function pre and post VPS placement?

The Line numbering restarted just before the discussions, please correct to better review process.

Table 4. please adjust the layout of the table so the word “meningoencephalitis” could be not discontinued

Please revised the format of the References there are some misaligned ones.

Finally I suggested a modified title that for me is more clearer than the previous one, but please feel free to choose the one you like: “Evaluation of overshunting between low and medium pressure ventriculoperitoneal shunts in dogs with severe hydrocephalus using frameless stereotactic ventricular shunt placement.”

Reviewer 2 Report

Dear Authors, I believe the scientific content of this manuscript is worthly fo being considered for publication. However information provided are not well organised, and the whole manuscript is someways very confusing for the reader.

In first instance I would suggest to make the text more concise. Then a massive revision of the language is required to make it sound.

Reviewer 3 Report

This is an interesting manuscript analysing the overshunting for treating hidrocephalus with low or medium-pressure valve. The
manuscript theme is interesting, with clinical application and well written
. However, I have some concerns that requires to be addressed.

The main problem is the low number of cases in order to make a proper statistics to compare both groups; even so it is an usefull description of clinical cases. In any case, in my opinion, a brief description of the descriptive statistics and the statistics software that has been carried out should be included in Material and Methods.

In Table 3, the last column mixes the medical treatment with the evolution of one of the patients. It should be broken down into two different columns.

In the tables, combine the number of decimals, since in some cases the authors put one decimal and in other cases they put two.

For all these reasons, it seems appropriate to accept the manuscript with minor revisions.

Reviewer 4 Report

Dear authors, thank you for submitting this interesting paper. There are still a few changes to make and some issues need to be discussed. Please find my comments on the original file. All the best. 

Round 2

Reviewer 4 Report

There are still many spelling mistakes and the word order in sentences is wrong, which makes the paper difficult to read. I can highly recommend using an editing service provided by different journals to improve the article. In Materials and Methods I would add the cutoff pressures, when you chose the low or medium pressure valve. Fixation of the device is not mentiones. Postoperatively they received a CT not a radiograph - is that right? Line 112

Figure 4english

Figure 5 : when was the postoperative MRI done (days, hours post surgery?)

Figure 6, time from surgery should be noted on each figure

Line 334 to 340 there are too many brackets, and you say mean is like 102 +/- 20 min (79-117) what are the values in the second bracket?

ceftriaxone and baytril do not really cross the blood brain barrier so why did you choose them. 
